# Aerosol Inhalation of Chimpanzee Adenovirus Vectors (ChAd68) Expressing Ancestral or Omicron BA.1 Stabilized Pre–Fusion Spike Glycoproteins Protects Non–Human Primates against SARS-CoV-2 Infection

**DOI:** 10.3390/vaccines11091427

**Published:** 2023-08-28

**Authors:** Shen Wang, Mian Qin, Long Xu, Ting Mu, Ping Zhao, Bing Sun, Yue Wu, Lingli Song, Han Wu, Weicheng Wang, Xingwen Liu, Yanyan Li, Fengmei Yang, Ke Xu, Zhanlong He, Michel Klein, Ke Wu

**Affiliations:** 1Regulatory and Medical Affairs Department, Wuhan BravoVax Co., Ltd., Wuhan 430070, China; shen.wang@bravovax.com (S.W.); lingli.song@bravovax.com (L.S.); 2Project Management Department, Wuhan BravoVax Co., Ltd., Wuhan 430070, China; mian.qin@bravovax.com (M.Q.); long.xu@bravovax.com (L.X.); 3Innovative Discovery Department, Wuhan BravoVax Co., Ltd., Wuhan 430070, China; ting.mu@bravovax.com (T.M.); bing.sun@bravovax.com (B.S.); 4Test Development Department, Wuhan BravoVax Co., Ltd., Wuhan 430070, China; ping.zhao@bravovax.com (P.Z.); yue.wu@bravovax.com (Y.W.); 5Quality Control Department, Wuhan BravoVax Co., Ltd., Wuhan 430070, China; wuhanwh2012@sina.com; 6Pilot Production Department, Wuhan BravoVax Co., Ltd., Wuhan 430070, China; weicheng.wang@bravovax.com; 7Quality Assurance Department, Wuhan BravoVax Co., Ltd., Wuhan 430070, China; xingwen417@163.com; 8Institute of Medical Biology, Chinese Academy of Medical Sciences and Peking Union Medical College, Kunming 650000, China; lyy110719@imbcams.com.cn (Y.L.); hzl612@126.com (Z.H.); 9State Key Laboratory of Virology, College of Life Sciences, Wuhan University, Wuhan 430072, China; xuke03@whu.edu.cn; 10Executive Office, Wuhan BravoVax Co., Ltd., Wuhan 430070, China; michel.klein@umontreal.ca; 11Executive Office, Shanghai BravoBio Co., Ltd., Shanghai 200000, China

**Keywords:** SARS-CoV-2, vaccine, aerosol, chimpanzee adenovirus vector, heterologous

## Abstract

Current COVID-19 vaccines are effective countermeasures to control the SARS-CoV-2 virus pandemic by inducing systemic immune responses through intramuscular injection. However, respiratory mucosal immunization will be needed to elicit local sterilizing immunity to prevent virus replication in the nasopharynx, shedding, and transmission. In this study, we first compared the immunoprotective ability of a chimpanzee replication–deficient adenovirus–vectored COVID-19 vaccine expressing a stabilized pre–fusion spike glycoprotein from the ancestral SARS-CoV-2 strain Wuhan–Hu–1 (BV-AdCoV-1) administered through either aerosol inhalation, intranasal spray, or intramuscular injection in cynomolgus monkeys and rhesus macaques. Compared with intranasal administration, aerosol inhalation of BV-AdCoV-1 elicited stronger humoral and mucosal immunity that conferred excellent protection against SARS-CoV-2 infection in rhesus macaques. Importantly, aerosol inhalation induced immunity comparable to that obtained by intramuscular injection, although at a significantly lower dose. Furthermore, to address the problem of immune escape variants, we evaluated the merits of heterologous boosting with an adenovirus–based Omicron BA.1 vaccine (C68–COA04). Boosting rhesus macaques vaccinated with two doses of BV-AdCoV-1 with either the homologous or the heterologous C68–COA04 vector resulted in cross–neutralizing immunity against WT, Delta, and Omicron subvariants, including BA.4/5 stronger than that obtained by administering a bivalent BV-AdCoV-1/C68–COA04 vaccine. These results demonstrate that the administration of BV-AdCoV-1 or C68–COA04 via aerosol inhalation is a promising approach to prevent SARS-CoV-2 infection and transmission and curtail the pandemic spread.

## 1. Introduction

Since COVID-19′s emergence in 2019 and its subsequent global expansion, over 760 million people have been infected, resulting in more than 6.87 million deaths. The pandemic remains a public Health Emergency of International Concern (WHO), and the long–COVID syndrome is a matter of serious concern. There are currently over 200 COVID-19 vaccine candidates in the research and development pipeline to address the worldwide disease burden [1], and an increasing number of safe and efficacious vaccines have been approved for emergency use [2].

An ideal SARS-CoV-2 vaccine should elicit broad, potent, and long–lasting immunity to prevent virus replication in the nasopharynx, shedding, and transmission. Various vaccine approaches have been successfully used to combat the pandemic, and new innovative strategies against SARS-CoV-2 and emerging variants are under active development [3]. Among them, recombinant adenoviruses (AdVs) are a promising platform for SARS-CoV-2 vaccine development due to their mature manufacturing process, high gene–transfer potential, and immunogenicity [4]. Their safety has been an asset in the development of several prophylactic vaccines and immunotherapeutics [5]. However, the high seroprevalence of human adenovirus vectors and pre–existing immunity, in particular against adenovirus 5 in the population, are potential limitations to their clinical application [6]. Therefore, the selection of adenoviral vectors from non–human primates, such as chimpanzees and gorillas, should circumvent interference from pre–existing anti–human adenovirus immunity.

Currently, SARS-CoV-2 vaccines are primarily administered intramuscularly, which triggers strong protective humoral and cell–mediated immune responses [7]. However, COVID-19 is a mucosally transmitted disease. SARS-CoV-2 infects and persists in the upper respiratory tract, where the nasal epithelium contains the highest concentration of the angiotensin–converting enzyme 2 (ACE2) receptor that promotes virus replication [8]. Conventional intramuscular immunization reduces the severity of disease and symptomatic cases, but it does not elicit secretory IgA (sIgA) responses at mucosal sites and, thus, does not provide significant protection against SARS-CoV-2 infection, shedding, and transmission [9].

This is why current efforts focus on the development of COVID-19 vaccines that can be directly administered to the mucosa of both the upper and lower respiratory tracts to induce sIgA, long–term lung resident memory T cells (T_RM_), and sterilizing immunity [10]. Pre–clinical and clinical studies have shown that intranasal (IN) vaccination produces strong local mucosal immunity as well as systemic protection comparable to intramuscular immunization [10]. Various IN vaccines against SARS-CoV-2 are under intensive investigation, with 12 candidates reaching clinical trials at different stages [10]. Furthermore, individuals with high levels of wild–type spike–specific mucosal IgA were shown to be at a lesser risk of subsequent Omicron breakthrough infection [11], and broadly neutralizing IgA antibodies elicited by mucosal tissues from Wuhan COVID-19 convalescent patients were found to potently neutralize Omicron BA.1 and BA.2 infections [12]. 

Although intranasal administration of influenza vaccines has been recommended, it is worth noting that the vast majority of vaccine droplets land in the anterior nose, with only a trace of them landing in the middle turbinate region [13]. Viral shedding and aerosols in the upper airways are the key contributors to viral spread [14]. An intranasal vaccination strategy, however, is not optimal to prevent SARS-CoV-2′s airborne transmission. Indeed, Altimmune announced that the development of its intranasal COVID-19 vaccine (AdCOVID) was terminated due to poor results from its phase I clinical trial [15]. Phase I trial data published by Oxford–AstraZeneca revealed that intranasal ChAdOx1 administration induced neither a consistent mucosal immune response nor a robust systemic response [16]. Therefore, direct pulmonary immunization using aerosol–generating technologies is potentially a more effective strategy to elicit immunity in both the upper and lower respiratory tracts. Aerosolization for pulmonary immunization not only eliminates the need for needles [17], but it is also superior to intramuscular injection [11] and intranasal administration [12] to protect against a variety of respiratory pathogen infections at their sites of entry [18]. An aerosolized human adenovirus 5–based COVID-19 vaccine elicited robust systemic and mucosal immunity against SARS-CoV-2 variants in rhesus macaques [19]. Preliminary results from recent phase I trials showed that two doses of the aerosolized adenovirus type–5 vector–based COVID-19 vaccine elicited neutralizing antibody responses similar to those obtained with one dose of intramuscular injection [15]. Another aerosol–inhaled COVID-19 vaccine developed by McMaster University that provides robust protection against both the ancestral strain and variants of SARS-CoV-2 has also moved to clinical trials (NCT05094609) [20].

There is compelling evidence that the protective immunity induced by two doses of current COVID-19 vaccines rapidly wanes over time, in particular in the elderly and vulnerable individuals with underlying medical conditions [21]. This situation is compounded by the emergence of the highly transmissible SARS-CoV-2 Omicron sublineages that escape vaccine–induced neutralizing immunity [22]. Booster vaccination with a third or fourth dose of a homologous or heterologous vaccine has been recommended to recall protective immunity in previously vaccinated individuals and broaden the spectrum of cross–neutralizing antibodies against variants of concern (VOCs) [23]. In particular, heterologous boosting with an inhaled adenovirus–based COVID-19 vaccine of subjects previously immunized with an inactivated vaccine produced high cross–neutralizing antibody titers against Omicron subvariants [24].

To broaden the efficacy of our SARS-CoV-2 vaccines, we engineered two chimpanzee adenovirus vectors (ChAd) expressing a stabilized SARS-CoV-2 prefusion spike glycoprotein either from the Wuhan–Hu–1 (BV-AdCoV-1) or the Omicron BA.1 (C68–COA04) strain. We first compared the humoral and mucosal responses to BV-AdCoV-1 in cynomolgus monkeys using different vaccination modalities, then in rhesus macaques using different vaccine doses. We next investigated whether homologous boosting with a monovalent BV-AdCoV-1 or heterologous boosting with the Omicron BA.1 vector (C68–COA04) could optimize cross–protective immunity against WT, Delta, and Omicron BA.1. BA.2 and BA.4/5 strains in BV-AdCoV-1–vaccinated rhesus macaques compared with a bivalent (BV-AdCoV-1/C68–COA04) booster vaccination. Our findings show that the administration of an aerosolized boosting with either BV-AdCoV-1 or C68–COA04 has the potential to optimize and broaden cross–neutralizing antibody responses against SARS-CoV-2 variants of concern, including Omicron sublineages.

## 2. Materials and Methods

### 2.1. Cells

African green monkey kidney cells (Vero) and human embryonic kidney cells (HEK293A cells) were purchased from American Type Culture Collection (ATCC, Rocville, MD, USA). Both cell lines were maintained in Dulbecco’s modified Eagle’s medium (DMEM, Gibco, Carlsbad, CA, USA) supplemented with 5% fetal bovine serum (FBS) (Gibco, Carlsbad, CA, USA) and 1% penicillin/streptomycin (Gibco, Carlsbad, CA, USA). Cells were cultured at 37 °C in a humid atmosphere incubator with 5% CO_2_.

### 2.2. Animals

Eight male cynomolgus monkeys (weighing 2.7–3.8 kg) were purchased from Guangxi XiongSen Experimental Animal Breeding Primate Development Co., Ltd., Guigang, China, and the immunogenicity study was conducted in JOINN Laboratories (Suzhou) Co., Ltd., Suzhou, China. A total of 30 rhesus macaques (females/males ratio of 1:1, weighing 3.0–6.0 kg) were purchased from Suzhou Xishan Zhongke Laboratory Animal Co., Ltd., Suzhou, China, and the immunogenicity studies were conducted in Suzhou Guochen BioTek Co., Ltd., Suzhou, China. All animals were housed individually in a 12 h light and dark cycle at 18–26 °C. Water and food were provided daily in sufficient quantities.

### 2.3. Ethic Statement

The vaccination protocol for the eight cynomolgus monkeys was approved by the Animal Ethics Committee (No. ACU21–1193) of JOINN Laboratories (Suzhou) Co., Ltd., Suzhou, China.

The vaccination protocol for the 30 rhesus macaques was approved by the Animal Ethics Committee (No. IACUC–21061501) of Suzhou Guochen BioTek Co., Ltd., Suzhou, China.

The vaccination protocol for 18 rhesus macaques was approved by the Animal Ethics Committee (No. IACUC–22042001) of Suzhou Guochen BioTek Co., Ltd., Suzhou, China.

The challenge protocol for 12 rhesus macaques was approved by the Institutional Animal Care and Use Committee of the Institute of Medical Biology, Chinese Academy of Medicine Sciences & Peking Union Medical College, Kunming, China (Approval Number: DWSP202107020).

### 2.4. Construction of Chimpanzee Adenovirus Vectors

#### 2.4.1. Generation of BV-AdCoV-1

The BV-AdCoV-1 vector expressing the ancestral Wuhan Hu–1 prefusion spike gene was constructed as previously described [25] and shown to protect golden Syrian hamsters against SARS-CoV-2 infection without disease enhancement post–challenge. The particle size distribution of aerosols generated by nebulizing BV-AdCoV-1 was determined using a Winner 311XP laser light scattering particle analyzer (Jinan, China).

#### 2.4.2. Generation of C68–COA04 (Omicron BA.1)

The COA04 gene cassette contains a 2P–based prefusion spike gene of the Omicron BA.1 variant (GenBank ID: UFO69279.1). As for the construction of BV-AdCoV-1, the spike gene was modified to (1) encode mutations of amino acids 982/983 to prolines; (2) substitute the furin cleavage site at residues 679–68 aa with GlySerAlaSer; (3) replace the original signal peptide (1–12 amino acids) with the Japanese encephalitis virus envelope signal peptide (Js SP); (4) delete the gene segments coding for the transmembrane and intracellular domains (1205–1270 aa) of the spike protein; (5) add the trimerization sequence from T4 phage fibritin to the C–terminus of the spike extracellular domain to stabilize the trimer.

The codon–optimized prefusion spike gene cassette was synthesized by Genecreate (Wuhan, China) and inserted into the chimpanzee AdC68 vector (Suzhou Xiangyi Biotechnology Co., Ltd., Suzhou, China) to yield the pAdC68–COA04 plasmid. After linearization, pAdC68–COA04 was transfected into 85% confluent HEK293A cells to rescue the recombinant adenovirus. After several amplifications of the rescued adenovirus in HEK293A cells, the virus was purified by anion–exchange and gel–filtration chromatography. The purified virus was named C68–COA04. Virus particle numbers and virus titers were determined by UV_260_ and ICC_50_ methods, respectively.

After virus propagation, the purified Omicron pre–S protein was analyzed by SDS–PAGE under reducing conditions followed by Western blotting using a monoclonal anti–Omicron virus spike antibody (Sino Biological Inc., Beijing, China, Cat 40592–MM117: 1:500). Protein bands were detected using Tanon 5200 Chemiluminescent Imaging System (Tanon, Shanghai, China).

The presence of replication–competent adenoviruses (RCAs) was assessed after three passages of the recombinant chimpanzee adenovirus vectors in A549 cells that do not express E1. The particle size distribution of aerosols generated by nebulizing C68–COA04 was determined using a Winner 311XP laser light scattering particle analyzer (Jinan, China).

### 2.5. Immunogenicity Study in Cynomolgus Monkeys

To compare the immune responses induced by aerosol or intranasal vaccine administration, eight male cynomolgus monkeys were divided into 2 groups (*n* = 4). Animals in group 1 received an intranasal dose of BV-AdCoV-1 (5 × 10^10^ VP/dose), and animals in group 2 were immunized by aerosol inhalation (IH) of BV-AdCoV-1 (5 × 10^10^ VP/dose). All monkeys were vaccinated using a two–dose regimen on day 0 (D0) and day 28 (D28). Aerosol inhalation devices were provided by Aerogen Ltd., Galway, Ireland, and intranasal devices by Wuxi NEST Biotechnology Co., Ltd., Wuxi, China.

The sera were collected to measure serum binding and neutralizing antibody titers on days –1 (D–1), 14 (D14), 26 (D26), 42 (D42), and 56 (D56). Bronchoalveolar lavage fluids (BALFs) and nasal lavage fluids (NLFs) were also collected to determine binding anti–wild type spike IgA antibody titers. Briefly, a catheter was introduced into the trachea after the animal was sacrificed, and the catheter and trachea were then stitched together. With a syringe, sodium chloride was progressively delivered to the lungs at a volume of 10–15 mL/kg. The alveoli were lavaged 3 times for approximately 10 s. Then, BALFs were collected and stored at −80 °C for subsequent assays.

To collect NLFs, 1 mL of 0.9% sterile sodium chloride was injected with a syringe into one nostril while pinching the other one. The fluid was aspirated back into the syringe. The procedure was repeated 3–4 times. The collected fluid was then used to lavage the other nostril. Finally, the NLFs were collected into a sterile container and stored at −80 °C for subsequent assays.

### 2.6. Immunogenicity Study Design in Rhesus Macaques

#### 2.6.1. Vaccination and Challenge

Thirty rhesus macaques were randomly assigned to six different vaccination regimens: a negative control group (NC) (0.9% sterile sodium chloride (Saline), aerosol inhalation, *n* = 6), group 1 (0.6 × 10^10^ VP/dose, aerosol inhalation, *n* = 4), group 2 (2 × 10^10^ VP/dose, aerosol inhalation, *n* = 4), group 3 (5 × 10^10^ VP/dose, aerosol inhalation, *n* = 6), group 4 (10 × 10^10^ VP/dose, aerosol inhalation, *n* = 6), and group 5 (5 × 10^10^ VP/dose, intramuscular administration, *n* = 4). Animals in each group were half males and half females. The details about the grouping and vaccination schedule are presented in Appendix A. Animals were all vaccinated with BV-AdCoV-1 using a two–dose regimen on day 0 (D0) and day 28 (D28). Blood samples were collected on D–1, D14, and D26 (two days before boost 1) and D42 (14 days after boost 1). Serum binding and neutralizing antibody titers were measured at each time point. Aerosol inhalation devices were provided by Aerogen Ltd. Galway, Ireland.

The animals in the NC group and group 3 (twelve in total) were transferred to the Institute of Medical Biology, Chinese Academy of Medicine Sciences & Peking Union Medical College. All experiments with live SARS-CoV-2 in NHPs were performed in an ABSL–3 facility. On D53, the 12 animals in the NC group and group 3 were challenged and referred to as the infected and vaccinated groups, respectively. Briefly, the animals were administered 1 × 10^6^ PFUs in 1 mL (500 μL intranasally and 500 μL intratracheally) of the SARS-CoV-2 GD108# strain, which is similar to the Wuhan Hu–1 strain. Weight and body temperature were recorded daily. The nasal, throat, and anal swabs were collected every two days for RT–qPCR analysis to determine the viral load. All animals were sacrificed 7 days post–challenge. The lungs were collected for Hematoxylin–Eosin (HE) staining and histopathology analysis. Additionally, selected tissues were analyzed for viral loads, and levels of cytokines were measured in sera and lung homogenates.

#### 2.6.2. Durability of Antibody Response and Sequential Immunization

The remaining 18 animals were periodically monitored to explore the durability of the antibody response until D249. After 8 months, the 18 rhesus macaques were regrouped into three groups, among which binding antibody titers did not differ significantly: Group 1: six animals received a second booster of C68–COA04 vaccine (5 × 10^10^ VPs) on D284; Group 2: six animals received BV-AdCoV-1 (5 × 10^10^ VPs); and Group 3: six animals received a combination of C68–COA04 (5 × 10^10^ VPs)/BV-AdCoV-1 (5 × 10^10^ VPs) vaccine. Inhalation devices were provided by Dongguan Aidisy Machinery and Electronic Equipment Co., Ltd., Dongguan, China. Sera were collected on D–1, D7, D14, D28, D56, and D84 to measure binding and neutralizing antibody titers against SARS-CoV-2 variants of concern (VOCs).

### 2.7. Enzyme–Linked Immunosorbent Assay (ELISA)

Immune sera, BALFs, or NLFs were analyzed by indirect ELISA for binding IgG and IgA titers as previously described [26]. Briefly, 96–well plates (Cat#3590; Corning, New York, NY, USA) were coated over–night at 4 °C with 2 µg/mL of either the SARS-CoV-2 Spike, S1, or RBD protein (Genscript Biotech Corporation, Nanjing, China) from the ancestral virus or the Omicron BA.1 strain (Sino Biological Inc., Beijing, China) in 0.05 M carbonate–bicarbonate buffer (pH 9.6), before blockade with 200 µL of blocking buffer (PBST containing 0.05% Tween–20 and 10% skimmed milk) at 37 °C for 2 h. The sera at a starting dilution fold of 1:100 and BALFs and NLFs at a starting dilution fold of 1:4 were 2–fold serially diluted with blocking buffer. Then, 100 μL of serially diluted samples were added to the wells, and plates were further incubated for an hour at 37 °C. After washing the plates with PBST three times, horseradish peroxidase (HRP)–conjugated goat anti–human IgG antibody (Invitrogen, cat# A18805; 1:10000) or (HRP)–conjugated goat anti–human IgA antibody (Invitrogen, cat# A18781, 1:1000) was added to the wells, and the plates were further incubated for an additional 1 h at 37 °C. After three washes with PBST, 100 μL/well of a 3, 3′, 5, 5′–Tetramethylbenzidine (TMB) solution was added. The plates were incubated in the dark for 15 min at room temperature, and reactions were terminated by adding 100 μL/well of 1 M H_2_SO_4_ before the absorbance values were read at 450 nm. Pre–immune serum (100–fold diluted) was used as the negative control, and an anti–SARS-CoV-2 serum prepared by BravoVax Biotech Co. Ltd., Wuhan (China) was used as a positive control. The cutoff value was calculated as being 2.1 times the mean OD_450_ values obtained for samples from the negative control. The endpoint titer was calculated as the highest sample dilution at which the OD_450_ value was equal to or greater than the cutoff value.

### 2.8. Enzyme–Linked Immunospot (ELISpot) Assay

To evaluate the T–cell responses after the second boost with either C68–COA04, BV-AdCoV-1, or the C68–COA04/BV-AdCoV-1 combination vaccine, blood samples from the 18 rhesus macaques were collected on D283. Peripheral blood mononuclear cells (PBMCs) were separated by Ficoll–Hypaque density gradient centrifugation. The numbers of PBMCs secreting IFN–γ and IL–4 were determined using IFN–γ and IL–4 ELISpot kits (3421M–4AST–2, 3410–2APW–2, Mabtech, Sweden) according to the manufacturer’s instructions. Briefly, 50 μL of PBMCs (5 × 10^6^ cells/mL) were added in duplicate to ELISpot plates pre–coated with anti–IFN–γ or anti–IL–4 antibodies. A Wuhan–Hu–1 or Omicron spike peptide pool (15–mers overlapping by 11–amino acid residues at a final concentration of 2 μg/mL per peptide) (GenScript Inc., Nanjing, China) was used to re–stimulate immune cells. An anti–CD3 monoclonal antibody (1:1000, Mabtech, Nacka Strand, Sweden) served as a positive control, and an FBS–free medium as a negative control. Cells were incubated at 37 °C in 5% CO_2_ for 48 h, then the cell suspension was discarded, and 100 μL of either anti–IFN–γ antibody conjugated to Alkaline phosphatase (ALP) (7–B6–1–ALP, 1:1000, Mabtech, Nacka Strand, Sweden) or anti–IL4 antibody conjugated to ALP (IL4–Ⅱ–ALP, 1:300, Mabtech, Sweden) was added to the plates. The plates were further incubated at room temperature for 2 h. The spots were visualized by adding BCIP/NBT–plus substrate (Cat#3650–10, Mabtech, Nacka Strand, Sweden) and counted using an automated ELISpot reader (Mabtech, Nacka Strand, Sweden).

### 2.9. Luciferase–Based Pseudovirus Neutralization Assay

Sera were inactivated for 30 min at 56 °C, then 4–fold serially diluted from a starting dilution fold of 1:20 with cell culture medium. 100 μL of diluted sera were incubated with 50 μL of luciferase–expressing SARS-CoV-2 pseudoviruses (1.3 × 10^4^ CCID_50_/mL) for one hour at 37 °C, and then the mixtures were distributed to the plates (150 μL/well). Vero cells (2 × 10^4^ in 100 μL) were then added and incubated at 37 °C for another 20–28 h. The cells were lysed, and the luciferase activity was measured by the Britelite Plus Reporter Gene Assay System (PerkinElmer, Waltham, MA, USA). The relative light unit (RLU) values were read using a chemiluminescence detector (EnSight, PerkinElmer, Waltham, MA, USA). The IC_50_ neutralizing antibody titer was calculated by the Reed–Muench method. Each serum sample was assayed at six dilutions in duplicates. Ancestral SARS-CoV-2 (WT), Delta, and Beta pseudoviruses were purchased from Beijing YunLing Biotechnology Co., Ltd., Beijing, China. Ancestral SARS-CoV-2 (WT), Delta, and Omicron subvariants (BA.1, BA.2, and BA.4/BA.5) pseudoviruses used in the Omicron vector immunogenicity study were donated by Wuhan University, Wuhan, China. The first WHO International Standard for anti–SARS-CoV-2 immunoglobulin (NIBSC, 20/268, high 20/150) was used to determine neutralizing antibody titers.

### 2.10. Cytopathic Effect (CPE)–Based Micro–Neutralization Assay

Sera were inactivated for 30 min at 56 °C. Based on virus–induced cytopathic effects, a microneutralization (MN) assay was used to assess the ability of macaques’ immune sera to neutralize authentic SARS-CoV-2. The inactivated sera at a starting dilution of 1:4 were 3–fold serially diluted and added to the 96–well plate pre–filled with 100 CCID_50_ in 50 μL per well of SARS-CoV-2 GD108# strain. After incubating the plates at 37 °C for 2 h, 100 μL of 1 × 10^5^ /mL Vero cell suspension (1 × 10^4^ in 100 μL) was added to the wells. The plates were then incubated for 5 days at 37 °C in 5% CO_2_. The cytopathic effects were then observed and recorded. The assays were performed in duplicates. The neutralizing antibody titers were defined as the reciprocal of the highest serum dilution at which a 50% reduction in CPE was observed.

### 2.11. Tissue Homogenization

Monkey tissues from the lungs were collected and cut into small pieces (~1–2 g). After being washed twice with cold PBS, the tissues were homogenized in 20 volumes of cold PBS using a glass homogenizer (Bead Ruptor 24 Elite, OMNI, Kennesaw, GA, USA). The homogenates were centrifuged at 7000× *g* for 10 min, and the supernatants were collected and stored at −80 °C for cytokine assays.

### 2.12. RT–qPCR

The viral load was measured by real–time quantitative PCR (RT–qPCR) after reverse transcription by targeting a region on the E gene of SARS-CoV-2 (E–Sarbeco, Sangon Biotech, Shanghai, China). Primers and probe sequences are listed in Appendix A. Briefly, total RNA was extracted from 0.1 g of the homogenized lung, nasal mucosa, trachea, bronchus, lung lymph nodes, neck lymph nodes, spleen, brain, heart, liver, kidney, intestine lymph nodes, and ovaries/testes tissues, as well as from nasal, throat and anal swabs of rhesus monkeys using the TaKaRa MiniBEST Viral RNA/DNA Extraction Kit Ver.5.0 (TaKaRa, Shiga, Japan). Gene expression was quantitatively assessed using the CFX96 Touch Real–Time P PCR Detection System (Bio–Rad, Hercules, CA, USA). The virus abundance was determined using the 2^−ΔΔCt^ method [27]. The data were normalized against the γ–actin house–keeping gene.

### 2.13. Cytokine Analyses

The cytokine levels in immune sera and lung homogenates were determined using the Non–Human Primate Cytokine and Chemokine–30 Plex Procartaplex Panel (Cat# EPX010–40420–901, Thermo Fisher Scientific, Carlsbad, CA, USA) according to the manufacturer’s protocol. In short, lung tissues were mechanically homogenized and centrifuged at 16,000× *g* for 10 min at 4 °C. The supernatants of the tissue homogenates were collected and diluted to 10 mg protein/mL with PBS. Then, 25 µL of diluted lung homogenates or sera and 25 µL of Universal Assay buffer were added to wells of a 96–well plate containing magnetic beads coated with different anti–cytokine capture antibodies. The plates were sealed after incubation on a shaker at 500 rpm for 60 min at room temperature in the dark. The beads were washed by putting the 96–well plate on a flat magnet for 30 s, after which the fluid was discarded by using an automated plate washer. The magnet was removed, and the beads were resuspended in 25 µL of a mixture of biotinylated detection antibodies. After washing the plates twice, 50 μL of streptavidin–R–phycoerythrin solution was added to each well for another 30 min incubation at room temperature. Beads were washed twice, and 20 μL of Reading Buffer was added to each well. After a 5 min incubation, the plates were read using the Luminex 100/200 Magpix system (Luminex Corporation, Austin, TX, USA).

### 2.14. Hematoxylin–Eosin (HE) Staining and Histopathology Scoring

Lung tissues from the left upper lobe, left lower lobe, right upper lobe, and right lower lobe were collected for HE histopathology analysis. Sections of 5 μm were cut from paraffin–embedded tissues, deparaffinized in xylene, passed through 100% ethanol, and rehydrated in tap water. Samples were stained with hematoxylin, decolorized with 0.125% HCl/70% ethanol, blued in blue returning liquid (Servicebio Technology Co., Ltd., Wuhan, China), counterstained with eosin, dehydrated, and mounted in Micromount (Servicebio Technology Co., Ltd., Wuhan, China) cover slipping medium at room temperature.

The lung lobes were used for pathological analyses and scored semi–quantitatively as Grade 0 (Within normal limits); Grade 1 (Minimal); Grade 2 (Slight), Grade 3 (Moderate), Grade 4 (Significant), and Grade 5 (Severe) [28]. Scoring was performed based on the number of lung lobes affected and the degrees of inflammatory and hemorrhagic lesions. An average lung lobe score was calculated by combining scores from each criterion. The digital images of HE–stained slides were examined under a microscope (Nikon Eclipse E100, Tokyo, Japan).

### 2.15. Chimpanzee Adenovirus Vector Neutralization Assay

Serum samples were serially diluted and incubated with an AdC68 vector expressing the green fluorescence protein (rAdC68–GFP, 1.2 × 10^5^ TCID_50_/well) for 1 h at 37 °C. The mixtures were inoculated to HEK293A cells (2 × 10^4^ cells/well). After incubation at 37 °C in 5% CO_2_ for 48 h, the fluorescent spots were counted under a fluorescence microscope (Olympus BX–60) with a 450 nm excitation. The half–maximal inhibitory concentration (IC_50_) values were calculated according to the Reed and Muench method [29].

### 2.16. Statistical Analysis

The data expressed as means ± standard deviation (SD) were processed using GraphPad software 9.0 (GraphPad Software, San Diego, CA, USA). Differences were identified using two or three–way ANOVA and considered significant when *p* < 0.05. Correlations were assessed by a non–parametric Spearman correlation test.

## 3. Results

### 3.1. Characterization of Recombinant Chimpanzee AdC68–Vectored Vaccines

The BV-AdCoV-1 and C68–COA04 vectors were both engineered to express stabilized trimeric pre–fusion spike proteins from the Wuhan–Hu–1 and Omicron BA.1 strains, respectively. Both genes were modified to code for a spike with a JEV signal peptide, a deletion of the transmembrane and the cytoplasmic domains, two–proline stabilizing substitutions (S–2P), a mutation of the furin cleavage site, and a C–terminal trimerization domain (Figure 1A,B). Figure 1C represents an illustration of a monkey inhaling through nebulization. The pre–S protein produced by the BV-AdCoV-1 vector was previously shown to be a non–covalent trimer [30]. Likewise, the stabilized Omicron pre–S protein was found to be a 250 kDa monomer by Western blotting analysis under reducing conditions (Figure 1D). The inhaled particle sizes affect their deposition location in the lungs. Particle sizes within 1–5 μm are suitable for delivery to the smaller bronchioles and alveoli, indicating effective deep lung deposition and suitable pulmonary inhalation [31]. Particle sizes of BV-AdCoV-1 within 1–5 μm accounted for 61.43% (Figure 1E), while particle sizes of C68–COA04 within 1–5 μm accounted for 60.06% (Figure 1F) of aerosolized particles, indicating their suitability for pulmonary inhalation delivery.

### 3.2. Aerosol Inhalation of BV-AdCoV-1 Elicits Stronger Humoral Responses against SARS-CoV-2 Than Intranasal Administration

We previously reported that the intranasal administration of BV-AdCoV-1 induced strong humoral and cellular protective immunity in golden hamsters [25]. We turned to the non–human primate model to further compare the respective efficiencies of intranasal administration (IN) and aerosol inhalation (IH). To this end, a total of eight cynomolgus monkeys were vaccinated with BV-AdCoV-1 to assess the humoral and mucosal responses induced using the intranasal route (Group 1) or nebulization inhalation (Group 2) (Figure 2A). Monkeys received two 5 × 10^10^ VPs–doses of BV-AdCoV-1 vaccine 28 days apart.

The monkeys were routinely monitored for body weight and temperature, and no significant differences were noticed between the immunization routes (Appendix A). To assess the dynamic changes in Spike (S)–, S1– and RBD–specific binding IgG antibody responses against WT strain, blood was collected on D–1, D14, D26, D42, and D56. Vaccinated monkeys in both groups developed binding IgG antibody responses against S, S1, and RBD from the WT strain that could be further boosted (Figure 2B–D). Furthermore, the anti–S1 and anti–RBD IgG titers generated in animals of the inhalation group were consistently and significantly higher than those in monkeys immunized intranasally on D42 (*p* < 0.05 vs. *p* < 0.05) and D56 (*p* < 0.05 vs. *p* < 0.01) (Figure 2C,D).

Both routes of immunization induced binding IgA responses against S from the WT strain in all animals (Figure 2E,F). On D56, serum the anti–S IgA titers (GMT 57) in monkeys immunized via the IH route were 9.5 times higher than those in animals of the IN group (Figure 2E). The binding IgA antibodies against WT stain could also be detected in NFLs and BALFs on D56 (Figure 2F). Aerosolization of BV-AdCoV-1 stimulated the production of S–specific IgA GMTs of 11 and 4 in BALFs and NFLs, respectively, whereas IN administration elicited an IgA response in NFLs only.

Neutralizing antibody responses were evaluated using pseudovirus neutralization assays. Neutralization of WT (GMT, 760) and Delta (GMT, 539) pseudoviruses was markedly higher in the sera of monkeys immunized via aerosol inhalation than in the sera of monkeys vaccinated intranasally (GMTs of 760 vs. 21 and 539 vs. 7, respectively). The inhalation again induced a 12–fold higher neutralizing GMT against the Beta pseudovirus in the IH group than in the IN group (24 vs. 2) (Figure 2G).

These results unambiguously indicated that BV-AdCoV-1 aerosol inhalation induced stronger neutralizing and mucosal responses than IN immunization.

### 3.3. Comparison of Immune Responses Induced by BV-AdCoV-1 Aerosol Inhalation versus Intramuscular Administration in Rhesus Macaques

We further assessed the advantages of IH over IM administration in rhesus macaques. To this end, 30 rhesus macaques were randomly divided into six groups (Figure 3A). The macaques from the other groups (1−4) were immunized with escalating doses of BV-AdCoV-1 via the IH route. Among them, six monkeys in group 3 received two BV-AdCoV-1 doses (5 × 10^10^ VP/dose) via the IH route, and rhesus macaques in group 5 four were administrated the same two doses but intramuscularly. Six monkeys administered with an equal volume of saline via the IH route served as the negative control group (NC group). During the following observation period, there were no abnormal changes in body weight and temperature in all groups (Appendix A).

Measurement of antibody titers in immune sera collected on D14, D26, and D42 post–vaccination showed that the levels of anti–S and anti–RBD IgG antibodies against WT strain slightly increased 14 days after the first dose and then continued to rise on D26 across the groups. A clear dose–response was observed in terms of kinetics and magnitude of the humoral response (Figure 3B,C). On D26, the GMTs of anti–S IgG antibodies in group 3 (5.0 × 10^10^ VP/dose) and group 4 (10.0 × 10^10^ VP/dose) were 1796 and 1131, respectively, and the GMTs of RBD IgG antibodies against WT strain were 4525 and 2540, respectively, indicating that the humoral response had reached a plateau at the 5 × 10^10^ VP/dose. Boosting on D28 further increased the binding IgG responses to S and RBD against the WT strain (Figure 3B,C). The highest anti–S (GMT 14368) and anti–RBD (GMT 20319) IgG antibody titers were obtained with two IH administrations of a 5 × 10^10^ VP/dose of BV-AdCoV-1 on D42, and these titers were significantly higher than those induced by two IM administrations of the same vector dose (anti–S IgG GMT = 3200, anti–RBD IgG GMT = 5382, *p* < 0.05). These data confirmed that the aerosolization of BV-AdCoV-1 at a 5.0 × 10^10^ VP/dose only elicited a more robust immunity against SARS-CoV-2 than IM injection and thus would potentially allow vector sparing for mass immunization. In addition, monkeys vaccinated by aerosolization of BV-AdCoV-1 (5.0 × 10^10^ VP/dose) developed a serum RBD–specific IgA antibody response against WT on D26 (GMT 8) that was boosted by a second administration of vector to a GMT of 74, whereas anti–S IgA antibodies were detected only on D42 (GMT 24) (Appendix A).

We next characterized the ability of BV-AdCoV-1 to neutralize an ancestral SARS-CoV-2 (WT) pseudovirus. Compared with the negative control group, monkeys from Group 3 and Group 4 developed neutralizing antibody responses on D26 (*p* < 0.05) after primary vaccination that was further significantly increased after the booster dose (*p* < 0.05) (Figure 3D). On D42, the GMTs of pseudovirus–neutralizing antibodies across all the IH groups were dose–dependent and reached 162 (group 1, 0.6 × 10^10^ VP/dose), 433 (group 2, 2.0 × 10^10^ VP/dose), 920 (group 3, 5.0 × 10^10^ VP/dose), and 576 (group 4, 10.0 × 10^10^ VP/dose), respectively. Monkeys immunized via inhalation with 5.0 × 10^10^ VP/dose of BV-AdCoV-1 developed significantly higher neutralizing responses than the monkey vaccinated with the same dose of vector intramuscularly (GMT 920 vs. 220, *p* < 0.05). The neutralizing GMT in group 5 immunized IM with 5 × 10^10^ VPs per dose was similar to that observed in group 1, which had received two 0.6 × 10^10^ VPs doses only (GMT 162 vs. 220, *p* < 0.05) via inhalation, indicating that a low dose of aerosolized BV-AdCoV-1 could elicit a humoral response identical to that obtained with a 10–fold higher dose of vaccine injected intramuscularly.

Furthermore, we performed authentic SARS-CoV-2 neutralization assays with 2 × 10^4^ CCID_50_ of the ancestor strain to further compare the neutralizing potency of BV-AdCoV-1 when it was administered via either the IH or the IM route. On D26 after priming, BV-AdCoV-1 induced higher levels of neutralizing antibodies in IH–immunized groups 2, 3, and 4 compared with the control (NC) group and the IM–vaccinated group 5 (Figure 3E, *p* < 0.05). Neutralizing antibody titers across the five vaccinated groups were boosted to reach a peak on D42 (Figure 3E). Interestingly, immune sera from group 3 monkeys which received two 5 × 10^10^ VP/doses of BV-AdCoV-1, had the highest live SARS-CoV-2 neutralizing activity (GMT 266), comparable to that of the WHO Reference (High 20/150, NIBSC code: 20/268 with the live SARS-CoV-2 neutralizing titer of 256). IH immunization with two doses of 0.6 × 10^10^ BV-AdCoV-1 VPs only also resulted in the production of neutralizing antibody titers (GMT 78) that were comparable to those obtained with two intramuscular injections of a 10–fold higher vector dose (5 × 10^10^ VPs) (GMT 96, *p* < 0.05) (Figure 3E).

We also investigated whether immunization with BV-AdCoV-1 could induce cross–neutralizing antibodies against SARS-CoV-2 variants of concern (VOCs) (Figure 3F–H). The two–dose vaccination regimen resulted in good cross–neutralizing responses against the Delta pseudovirus (Figure 3H) but only one–log lower neutralizing antibody titers against the Beta strain (Figure 3G). On D42, GMTs against the Delta pseudovirus were 203, 212, 640, and 172 in IH–immunized groups 1–4, respectively, but only 35 in the IM–immunized group 5. At the identical dose of 5 × 10^10^ VPs, inhalation of BV-AdCoV-1 elicited significantly higher cross–neutralization responses against Delta (GMT 640) than IM administration (GMT 35) (*p* < 0.05) (Figure 3H). Cross–neutralizing antibody GMTs against the Beta pseudovirus were 16, 44, 42, and 21 in IH–immunized groups 1–4, respectively, but just 7 in the IM group (Figure 3G). These results confirmed that aerosol inhalation of BV-AdCoV-1 induced more robust cross–neutralizing antibody responses against the SARS-CoV-2 VOCs in rhesus macaques than IM injection.

Finally, we analyzed the correlation between the binding antibody response and the neutralizing response in the IH–vaccinated animals. A strong positive correlation (r = 0.9615) could be established between anti–S binding IgG titers and neutralizing antibodies titers against authentic SARS-CoV-2 (GD108# strain) (Figure 3I) as well as between anti–RBD binding IgG against WT strain and authentic neutralizing antibody titers (r = 9576) (Figure 3J). The pseudovirus–neutralizing antibody titers also correlated well with those measured in the authentic neutralization assay (r = 0.9515) (Figure 3K). Neutralizing antibody titers moderately (r = 0.5755) correlated with anti–RBD IgA titers (Figure 3L).

### 3.4. BV-AdCoV-1 Aerosol Inhalation Protects Rhesus Macaques against Live SARS-CoV-2Challenge

We next assessed the protective ability of BV-AdCoV-1 in rhesus macaques immunized twice with 5 × 10^10^ VPs by aerosol inhalation against live virus challenge. We chose the macaques from group 3 as they had developed the best humoral responses. Twenty–five days after the booster immunization, the monkeys were infected with 1 × 10^6^ pfu of SARS-CoV-2 (GD108#) intratracheally (0.5 mL) and intranasally (0.5 mL). The control group received saline and served as the infected group.

The body weight and temperature of the animals were monitored daily. No significant difference in monkeys’ pre–challenge and post–challenge weights was noticed. The pre–challenge and post–challenge temperatures also remained within the normal range (Figure 4A,B). After the sacrifice of the animals on day 7 post–challenge, the viral load was measured in lung tissues and several other selected tissues, including nasal mucosa, trachea, bronchi, neck lymph nodes, intestinal lymph nodes, brain, heart, liver, spleen, kidney, and ovaries/testes. As shown in Figure 4C, the vaccinated group exhibited an almost 3–log reduction in lung viral loads compared to the infected group. Viral loads of the lung inversely correlated with anti–S IgA antibody titers on D42 (r = −0.6667) (Figure 4D). The same inverse correlation was also observed between viral loads in the lungs and anti–RBD IgA antibody titers against WT strain on D42 (r = −0.6670) (Figure 4E) or with neutralizing antibody titers (r = −0.6523) (Figure 4F).

The lung viral RNA copies were below the limit of detection (LOD) in 4 out of 6 vaccinated animals. A reduction in viral RNA copies was also observed in nasal swabs from vaccinated animals compared to 7 dpi (Appendix A). Although not significant, slightly lower viral loads were also detected in the throat and anal swabs, as well as in some other tissues from the BV-AdCoV-1 group (Appendix A).

Cytokine levels in immune sera and lung homogenates were determined by Luminex analysis. There was a significant decrease (*p* < 0.05) in serum TNF–α level in infected rhesus macaques’ sera post–challenge, while there was also a significant decrease (*p* < 0.05) in serum TNF–α level in the BV-AdCoV-1 group, between pre–challenge and post–challenge. Except for TNF–α, there was not any significant difference between the pre– and post–challenge levels of other pro–inflammatory cytokines, including IL–2, IL–4, IL–5, IL–6, IL–10, IL–13, and INF–γ (Appendix AE). No increase in pro–inflammatory cytokines levels in lung homogenates of vaccinated animals was observed after the challenge (Appendix A). These results suggest that vaccination with BV-AdCoV-1 did not induce any enhanced disease.

Histopathological lesions in the upper left, upper right, lower left, and lower right lobes and bronchus were analyzed after HE staining. These lesions were characterized by inflammatory cell infiltrates, thickening of alveolar walls and septa, and hemorrhages, often with the presence of red blood cells in the alveolar space and bronchi. As shown in Figure 4G,H, inflammatory and pathological lesions in the lung tissues of immunized animals were reduced. The infected monkeys developed a mean interstitial pneumonia score of 1.625, while this score was only 0.854 in BV-AdCoV-1 vaccinated animals (Appendix A). The mean hemorrhage scores of 1.438 and 1.667 in infected and vaccinated monkeys were not significantly different (Appendix A). Overall, the mean lung histopathology score of BV-AdCoV-1 vaccinated animals was 2.520, while that of the infected group was 3.063 (Appendix A).

In summary, BV-AdCoV-1 aerosol inhalation protected against SARS-CoV-2infection without causing enhanced disease upon live virus challenge.

### 3.5. Heterologous Boosting with the Omicron BA.1–based C68–COA04 vaccine Enhances Cross–Neutralizing Antibody Responses against Wild–Type SARS-CoV-2, Delta, and Omicron Subvariants

Given the rapid expansion of the epidemic Omicron virus and its escape from previous vaccination against the ancestral strain, we explored whether boosting macaques immunized with BV-AdCoV-1 with a vector expressing the heterologous Omicron BA.1 pre–S protein would be more efficacious than a homologous boost at eliciting functional cross–neutralizing antibodies against SARS-CoV-2 VOCs, in particular Omicron subvariants (Figure 5A).

We first performed a longitudinal analysis to evaluate the durability of the humoral responses in the 18 BV–AdCo–V–1–immunized macaques from groups 1, 2, 4, and 5 that had not been sacrificed. As shown in Appendix A, anti–S and anti–RBD IgG antibody titers against the WT strain reached a peak on D42 in all vaccinated groups. After a decrease in IgG antibody titers four weeks after the first boost on D58, titer values plateaued for 3–4 months (from D42 to D140). The binding IgG antibody responses against WT strain in animals immunized with two low aerosol doses of 0.6 × 10^10^ VPs were comparable to those elicited by two IM injections of 5 × 10^10^ VPs. Anti–S and anti–RBD IgG antibody levels against WT strain progressively waned down to almost similar levels across the groups on D249 before animals were regrouped for further studies.

In parallel, we analyzed the cell–mediated immune responses on D283 after two doses of BV-AdCoV-1 by evaluating the numbers of IFN–γ/IL–4 secreting PBMCs after re–stimulation with either a WT or an Omicron spike peptide pool. Noticeably, in animals immunized intramuscularly, the number of IFN–γ–secreting cells re–stimulated with the WT spike peptide pool was four–fold higher than that obtained by re–stimulation with Omicron spike peptides (*p* < 0.05), suggesting that Th1 responses to WT spike were still higher than those to the Omicron spike. IL–4 ELISpot responses were low (0–28 spots/10^6^ cells) regardless of the type of PBMCs’ re–stimulation or the route of vaccine administration (Appendix A).

We then regrouped the 18 rhesus macaques into 3 groups of 6 animals that did not significantly differ in their levels of serum anti–S, anti–RBD, and anti–chimpanzee vector neutralizing antibody titers (Appendix A) to assess whether boosting with a heterologous Omicron spike could enhance pre–existing humoral immunity more efficiently than a second homologous booster. On D284, rhesus macaques were re–immunized with either the ancestral BV-AdCoV-1 vaccine (5 × 10^10^ VPs), the Omicron C68–COA04 vector (5 × 10^10^ VPs), or a combination of BV-AdCoV-1 (5 × 10^10^ VPs) and C68–COA04 (5 × 10^10^ VPs), respectively (Figure 5A).

The two monovalent vectors and the combination vaccine strongly boosted binding IgG and IgA responses against the WT or/and the Omicron spike, respectively (Figure 4B–E). Although the two monovalent and the bivalent vaccine elicited equivalent IgA titers against the WT spike antigen (Figure 5B), the BV-AdCoV-1 vector was significantly less potent than the other two vaccines at inducing binding IgA antibodies against the Omicron spike (*p* < 0.05, BV-AdCoV-1 vs. C68–COA04; *p* < 0.001, BV-AdCoV-1 vs. C68–COA04+BV-AdCoV-1) on D28 and D56 (Figure 5C). The monovalent C68–COA04 and the bivalent BV-AdCoV-1/C68–COA04 vaccines outperformed BV-AdCoV-1 at eliciting IgG antibody responses to both the WT and Omicron strains’ spikes on D56 (*p* < 0.01, BV-AdCoV-1 vs. C68–COA04+BV-AdCoV-1 or C68–COA04) and D84 (*p* < 0.05, BV-AdCoV-1 vs. C68–COA04+ BV-AdCoV-1, *p* < 0.01, BV-AdCoV-1 vs. C68–COA04) (Figure 5D,E).

Booster vaccination with the three types of vaccines markedly enhanced cross–neutralizing responses against the WT strain, Delta strain, and Omicron subvariants strains BA.1, BA.2, and BA.4/5 (Figure 5F–J). The monovalent C68–COA04 vaccine boost produced the highest levels of neutralizing antibodies against the WT strain (GMT 33432), followed by BV-AdCoV-1 alone (GMT, 11972) (Figure 5F). The strongest neutralizing response against the Delta strain (GMT 13772) was induced by BV-AdCoV-1 alone, followed by C68–COA04 (GMT, 3322) and then the bivalent vaccine (GMT 903) (Figure 5G). The highest neutralizing antibody titers against the Omicron BA.1 strain were elicited by the homologous monovalent C68–COA04 vaccine (GMT 702), whereas the bivalent vaccine and BV-AdCoV-1 elicited lower titers (GMTs 83 and 280, respectively) (Figure 5H). GMTs of neutralizing antibodies to the Omicron BA.2 strain in the C68–COA04 and bivalent vaccine groups were 1804 and 1774, respectively, nearly 30–fold higher than the GMT obtained with a homologous BV-AdCoV-1 booster dose (Figure 5I). However, the BV-AdCoV-1 boost elicited cross–neutralizing antibody responses against the Omicron BA.4/BA.5 strain (GMT, 1009) that were three–fold higher than those observed in the other two groups, although the difference was not statistically significant (*p* > 0.05) (Figure 5J). Taken together, these results demonstrate that a third dose vaccination schedule of monovalent C68–COA04 or BV-AdCoV-1 might equally optimize the breadth and potency of cross–neutralizing antibody responses against WT and Omicron subvariant strains, including the BA.1, BA.2, and to a lesser extent BA.4/5 sublineages.

## 4. Discussion

The objective of the study was to evaluate the efficacy of mucosal administration of optimized chimpanzee adenovirus vectors expressing prefusion–stabilized spike proteins at inducing immunoprotection against SARS-CoV-2 and broad cross–neutralization of variants of concern.

Simian adenovirus vectors are a promising vaccine platform [32] to induce sterilizing immunity against SARS-CoV-2, not only because they have a good Th1 immunogenicity profile and are easy to administer mucosally but also because they have a low seroprevalence in humans [33] and thus circumvent interference by pre–existing anti–human Ad5 immunity [34]. In addition, chimpanzee ChAd68–based vectors induce sustained immunity [35]. We selected the replication–deficient AdCh68 adenovirus vector, which is serologically and phylogenetically distinct from ChAd–Y25 used to engineer ChAdOx1 to express SARS-CoV-2 pre–fusion spike proteins that are more immunogenic than their native counterparts [36]. The two purified AdCh68–based vectors expressing either the Wuhan–Hu–1 or the Omicron BA.1 prefusion spikes used in this study were found to be free of RCA after three passages in A549 cells and thus are safe for clinical use. Both pre–S proteins secreted in the supernatants of vector–infected HEK293 cells formed non–covalent stabilized trimers as judged by Western blotting, size exclusion chromatography, and transmission electron microscopy.

Four adenovirus–based COVID-19 vaccines, Ad26.CoV2.S (Johnson & Johnson, New Brunswick, NJ, USA), Vaxzevria (ChAdOx1 nCoV–19, Oxford/AstraZeneca, Oxford, UK), Convidecia (Ad5–nCoV, CanSinoBIO, Tianjin, China), Sputnik V (Ad5/Ad26, Gamaleya Institute, Moscow, Russia) approved for human use are administered intramuscularly. Intranasal SARS-CoV-2 vaccines have been shown to reduce viral load and subsequent viral transmission and are currently under intensive development [10].

ChAd–based vaccines administered intranasally efficiently protected the respiratory tract of mice, rats, hamsters, and rabbits against SARS-CoV-2 infection [37]. Similar protective effects have also been observed in non–human primates. A single intranasal administration of a simian ChAd–36 vector expressing a pre–S protein resulted in the production of robust humoral and mucosal responses that prevented or limited SARS-CoV-2 infection in rhesus macaques [38]. The mucosal administration of a live–attenuated respiratory syncytial virus expressing a chimeric SARS-CoV-2 spike in African green monkeys elicited mucosal IgA in the nose, reduced virus shedding by more than 200–fold and protected animals against viral challenge [39]. However, the approved ChAdOx1 vaccine, which expresses a native spike and induces protective humoral and cellular responses when administered intramuscularly, failed to elicit substantial mucosal or systemic responses in human vaccinees when administered intranasally.

Several studies have established the superiority of intranasal immunization and nebulization inhalation over intramuscular injection in inducing protective immunity against SARS-CoV-2 in animal models [26,40]. However, intranasal vaccination can only deliver vaccines to the upper respiratory tract. In contrast, aerosol inhalation [10] efficiently delivers vaccines to both the upper and lower respiratory tracts and thus elicits stronger mucosal and systemic immune responses than intranasal immunization in animals and humans [21]. We first confirmed in cynomolgus macaques, which are a good surrogate for humans in inhalation studies [41], that aerosol inhalation outperformed vaccination by intranasal administration. No adverse events were observed upon vaccination. Immunization with the BV-AdCoV-1 vector (5 × 10^10^ VPs) expressing a stabilized trimeric pre–fusion spike from the ancestral Wuhan Hu–1 strain consistently induced 10– to 20–fold higher anti–spike, anti–S1, anti–RBD, and neutralizing antibody responses than intranasal administration. Nebulization inhalation also produced higher S– and RBD–specific serum monomeric mIgA responses and secretory sIgA in nasal and bronchoalveolar fluids than intranasal immunization, which failed to elicit detectable sIgA in nasal fluids under the test conditions used in this study. Kim et al. [42] reported that only intranasal or sublingual but not intramuscular immunization with an adenovirus 5 vector expressing S1–induced neutralizing IgA antibodies in BALFs. However, it was reported that serum anti–spike IgA levels in subjects immunized IM with the BNT162b2 mRNA vaccine correlated with anti–RBD and neutralization antibodies. Muller et al. [43] proposed that high serum IgA responses might be associated with more effective protection against breakthrough infections. Early SARS-CoV-2–specific humoral response in humans is dominated by IgA antibodies that greatly contribute to virus neutralization [17]. Dimeric and polymeric sIgA linked by the J piece predominate in the airways and play a major role in virus neutralization at the nasopharyngeal entry site [17]. Zhang et al. [44] reported that RBD–specific IgA in convalescent sera correlated with IgG responses and that if IgA monomers were less neutralizing than IgG antibodies, dimeric IgA in secretions were 15 times more potent neutralizing antibodies than their IgG counterparts. Human IgA dimers were also shown to efficiently neutralize Omicron BA.1 and BA.2 variants [45]. In the present study, we found that anti–RBD IgA moderately correlated (r = 0.5755) with virus neutralization and noticed a moderate inverse correlation (r = −0.6667, r = −0.6670) between lung viral load and serum anti–S/RBD IgA antibodies. We were not able to assess the relative percentages of monomeric versus dimeric sIgA in BALFs due to the lack of commercial anti–chimpanzee J pieces and anti–secretory component antibodies.

Pseudovirus cross–neutralizing antibody titers against the Delta strain were similar to those obtained for the ancestral strain and 77 times higher than those elicited by IN immunization. Meanwhile, the cross–neutralizing antibody GMT against the Beta strain was 12 times higher than that elicited by IN immunization. These results indicate that aerosol inhalation elicits stronger cross–neutralizing antibody responses and mucosal immunity than intranasal administration.

Aerosol inhalation was, therefore, used in rhesus monkeys to perform comparative immunogenicity studies with intramuscular vaccination. A dose–escalation study with BV-AdCoV-1 showed that two administrations of 5 × 10^10^ VPs were necessary and optimal to induce significantly higher spike– and RBD–specific IgG antibody levels than those elicited by intramuscular immunization. This two–dose regimen also produced serum–binding anti–S and anti–RBD IgA antibodies. Strong positive correlations were established between S– (r = 0.9615) or RBD–specific (r = 0.9576) IgG and neutralizing antibody titers, which are the best correlates of protection. The level of neutralizing antibodies was found to inversely correlate with the viral load (r = −0.6523). High positive correlations of neutralizing antibody titers with anti–spike and anti–RBD IgG antibodies and with ACE–2 binding inhibition were also observed in mRNA–vaccinees and infected patients. These immunological markers are thus good predictors of the protective potency of SARS-CoV-2 vaccines [46]. The 5 × 10^10^ dose of BV-AdCoV-1 elicited neutralization titers (GMT 266) comparable to those obtained for the WHO reference and in the range of those correlating with high protective efficacy, namely GMTs of 50 to 500 in rhesus macaques [47] and above GMT of 100 in humans [48]. Moreover, two 0.6 × 10^10^ VP doses only of inhaled vector were sufficient to generate pseudovirus neutralization antibody titers equivalent to those obtained with a 10–fold higher dose (5 × 10^10^ VPs) of the same vector injected intramuscularly, allowing for vaccine–sparing for mass immunization. Two 5 × 10^10^ VP doses of aerosolized BV-AdCoV-1 also elicited higher levels of cross–neutralizing antibodies against SARS-CoV-2 Delta and, to a lesser extent, against the Beta variant than IM administration. Furthermore, aerosolized immunization with the ChAd vector expressing prefusion spike protein markedly reduced the viral load in the lungs and nasal swabs and protected animals against pathology. Interestingly, although COVID-19 vaccines consistently induce low neutralizing antibody titers against the Beta strain, they can still protect rhesus macaques in a Beta variant challenge model [49].

T–cell adaptive immunity plays a critical role in virus clearance, reducing disease severity and long–term antiviral immunity. The induction of strong T–cell responses is a distinct feature of the ChAd vectored vaccines [50]. Vaccination with ChAdOx1 nCoV–19 promotes robust cellular immunity and the expansion of Th1 cells in macaques [51]. Similar responses were observed in humans vaccinated with the ChAdOx1 vaccine [43], and a polyfunctional Th1 response potentially resilient to spike point mutations was observed in AZD1222/ChAdOx1–vaccinated adults, whereas CD4^+^ Th2 responses were not detected [52]. In our study, BV-AdCoV-1 expressing prefusion spikes predominantly induced Th1 responses as judged by the high numbers of IFN–γ and very low numbers of IL–4–producing PBMCs after spike peptide re–stimulation. Such a response might contribute to protection in case of reduced neutralizing antibody titers or low antibody avidity. Although not observed so far with spike–based COVID-19 vaccines, immunopotentiation occurring in animals immunized with SARS–CoV–1 has been linked to the production of insufficient amounts of antibodies and a skewed immune response toward T–helper cell type 2 (Th2). The BV-AdCoV-1 vector, which elicits robust neutralizing antibodies and weak Th2 responses, did not cause vaccine–associated enhanced disease. Most pro–inflammatory cytokines were hardly detectable in the sera and lungs of vaccinated animals after viral challenge, and there was no increased secretion of TNF–α and IFN–γ in sera nor of IL–5 in the lungs compared to non–vaccinated monkeys.

The prime–boost schedules of COVID-19 vaccines have unambiguously demonstrated their safety, clinical effectiveness, and efficacy [25]. However, recent reports have consistently highlighted that immunity begins to decline 6–8 months after two vaccine doses [1], exposing vaccinees to re–infection or breakthrough infections, in particular by emerging SARS-CoV-2 variants. Adenovirus vectors are known to induce durable antibody responses and long–lasting T–cell immunity [53]. We found that six months after the second BV-AdCoV-1 dose, anti–RBD and anti–S antibody GMTs in 18 rhesus macaques from different immunization groups plateaued before waning down to lower levels. Since the emergence of the highly transmissible Omicron BA.1 in 2021, Omicron sub–lineages, such as BA.2 and BA.4/5, have successively rapidly spread worldwide and escaped persistent immunity in convalescent patients [54] as well as neutralization in fully vaccinated individuals [55]. Three exposures to SARS-CoV-2 spike by either infection or vaccination were found to elicit superior neutralizing activity to all variants of concern [56], and a third dose of the COVID-19 vaccine is now required to boost pre–established immune responses [23]. We indeed observed that a second homologous booster of BV-AdCoV-1 on D284 dramatically enhanced pre–existing mucosal and humoral immunity against SARS-CoV-2 WT and the Delta strain and that the efficacy of this second booster was not affected by anti–adenovirus vector immunity.

Several studies have revealed that boosting fully vaccinated animals or human subjects with a heterologous vaccine type was more efficient than administering a third dose of homologous immunogen [57]. Priming of macaques with a chimpanzee adenovirus vector ChAd–S expressing a full–length ancestral protein followed by an inactivated virus vaccine [49] significantly improved the immune responses and neutralized Omicron variants. Heterologous prime–boost immunization with ChAdOx1–nCoV and the mRNA BNT162b2 vaccine achieved very high neutralizing antibody titers and Th1 responses in humans [58]. Aerosol inhalation of a hAd5–vectored vaccine expressing the Wuha–Hu–1 spike generated stronger neutralizing responses against the D614, BA.2, and BA.5 strains in subjects having received an inactivated virus vaccine than homologous vaccination [37]. In the present study, we assessed the merits of a heterologous prime–boost schedule using the same ChAd68 vector backbone to express two divergent SARS-CoV-2 spikes. We tested whether the C68–COA04 vector expressing the Omicron BA.1 pre–S protein could enhance the pre–existing neutralizing antibody responses provided by BV-AdCoV-1 against the ancestral strain and SARS-CoV-2 VOCs in particular against the epidemic Omicron sublineages which have recently spread worldwide. A second boost with C68–COA04 induced IgA responses against WT and BA.1 spikes and significantly increased pre–existing binding IgG titers against both strains and neutralizing antibodies against the ancestral strain, more than the monovalent BV-AdCoV-1 alone, perhaps due to part to antigenic imprinting. The booster dose also dramatically enhanced neutralizing immunity against WT, Delta, and Omicron variants BA.1 and BA.2. The bivalent vaccine combining both vectors did not improve the immune responses obtained with its components. But interestingly, regardless of the vaccine used as a boost, the prime–boot regimens used in the study elicited stronger neutralizing antibody responses against BA.2 than against BA.1, which is in line with the results of a previous report that three doses of homologous mRNA vaccination or a heterologous adenovirus/mRNA schedule weakly neutralize the BA.1 sub–lineage compared to BA.2 [59]. The two monovalent and the bivalent vaccines elicited substantial cross–neutralizing antibody responses against Omicron BA.4/5 variants. Indeed, homologous booster with vector–based vaccines expressing the same antigen is less effective than heterologous booster with the same vector–based vaccines expressing different immunogens [60]. The heterologous immunization approach is thus a potentially efficacious approach to controlling infection from the Delta strain, the Omicron sublineages, and the hybrid “Deltacron” variant [60].

## 5. Conclusions

In summary, we found that aerosol inhalation of BV-AdCoV-1 induced significantly stronger neutralizing and secretory sIgA responses than intranasal administration and is superior to intramuscular injection. It also elicited a Th1 response and reduced nasal and lung viral loads while conferring immunoprotection against lung pathology without causing enhanced disease. Aerosolization of either the BV-AdCoV-1 or C68–COA04 monovalent vaccines as second homologous or heterologous boosters dramatically recalled residual cross–neutralizing responses against the Delta, BA.1. BA.2 and BA.4/5 variants of concern. These data suggest that aerosol inhalation is a promising platform to prevent infection, reduce shedding, and limit transmission and, thus, should become the preferred route to deliver stand–alone COVID-19 vaccines or to perform booster immunization.

## Figures and Tables

**Figure 1 vaccines-11-01427-f001:**
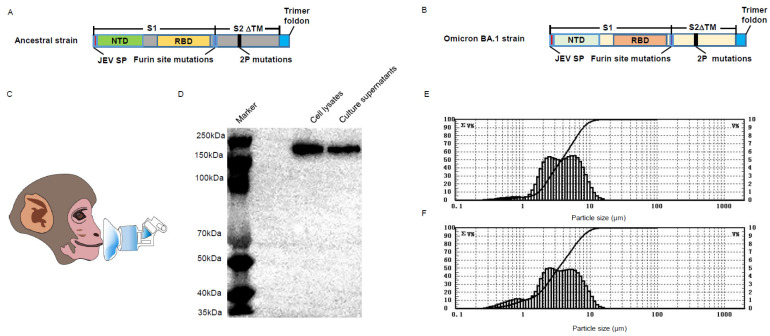
Characterization of the wild–type (WT) and Omicron BA.1 pre–S protein. (**A**) Design and construction of BV-AdCoV-1 vector expressing the pre–S protein. (**B**) Design and construction of C68–COA04 vector expressing the pre–S protein. (**C**) Drawing of the aerosol inhalation devices for the macaques. (**D**) Western blot analysis of the Omicron pre–S protein. (**E**) Nebulized aerosol size distribution of BV-AdCoV-1. (**F**) Nebulized aerosol size distribution of C68–COA04.

**Figure 2 vaccines-11-01427-f002:**
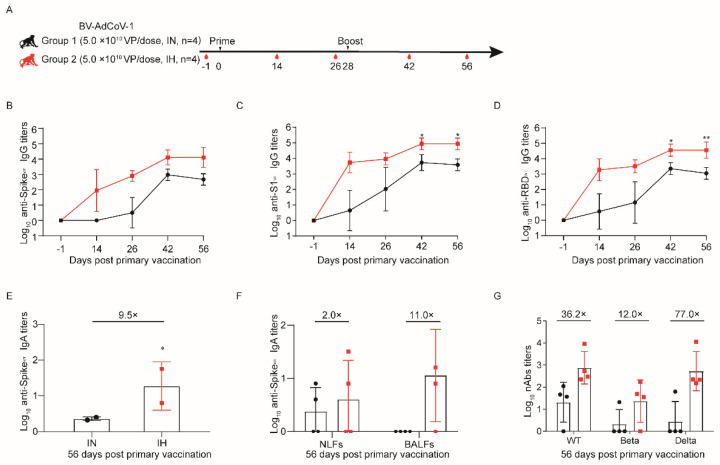
Aerosol inhalation elicits stronger protective immunity against SARS-CoV-2 than intranasal administration. Eight male cynomolgus monkeys (aged 3–4 years) were divided into two groups (*n* = 4). Group 1 was immunized with two 5.0 × 10^10^ VP doses by the intranasal route (IN), whereas Group 2 was immunized by inhalation (IH) with two 5 × 10^10^ VP/doses of BV-AdCoV-1. (**A**) Schedule of vaccine administration and blood collection. Dosing (black triangles) and terminal blood collections (red drop shapes) were carried out at the times. (**B**) Binding IgG antibody titers against S of wild–type (WT) SARS-CoV-2 were determined by indirect ELISA for sera collected on D–1, D14, D26, D42, and D56 after primary immunization. (**C**) Binding IgG antibody titers against S1 of WT SARS-CoV-2 were determined by indirect ELISA for sera collected on D–1, D14, D26, D42, and D56 after primary immunization. (**D**) Binding IgG antibody titers against RBD of WT SARS-CoV-2 were determined by indirect ELISA for sera collected on D–1, D14, D26, D42, and D56 after primary immunization. (**E**) Binding IgA antibody titers against S of WT SARS-CoV-2 in cynomolgus monkeys’ sera were determined by indirect ELISA on D56 after primary immunization. Data are presented as means ± SD. On D56, the IgA titers in the IH group were almost 9.5 times (9.5×) higher than that in the IN group. (**F**) Binding IgA titers against S of WT SARS-CoV-2 in nasal lavage fluids (NLFs) and bronchoalveolar lavage fluids (BALFs) of cynomolgus monkeys were determined by indirect ELISA on D56 after primary immunization. On D56, the IgA titers in the NLFs and BALFs of the IH group were almost 2.0 (2×) or 11.0 (11.0×) times higher than those in the IN group, respectively. (**G**) Pseudovirus neutralizing antibody titers against WT, Beta, and Delta strains in cynomolgus monkeys’ sera on D56 after the first immunization. On D56, the neutralizing antibody titers against pseudoviruses of SARS-CoV-2 Wuhan–Hu–1 (WT), Beta, and Delta in the IH group were 36.2 (36.2×), 12.0 (12.0×) and 77.0 (77.0×) times higher than those in the IN group, respectively. Data are presented as means ± SD. The lower limit of detection is an IgG titer of 100 in sera. When the value is below 100, it is calculated as 1. The lower limit of detection is an IgA titer of 20 in sera. When the value is below 20, it is calculated as 1. The lower limit of detection is an IgA titer of 4 in NLFs and BALFs Statistical analysis was performed using two–way ANOVA and Tukey’s multiple comparison test. * *p* < 0.05 and ** *p* < 0.01. IH: Aerosol inhalation. IN: intranasal administration.

**Figure 3 vaccines-11-01427-f003:**
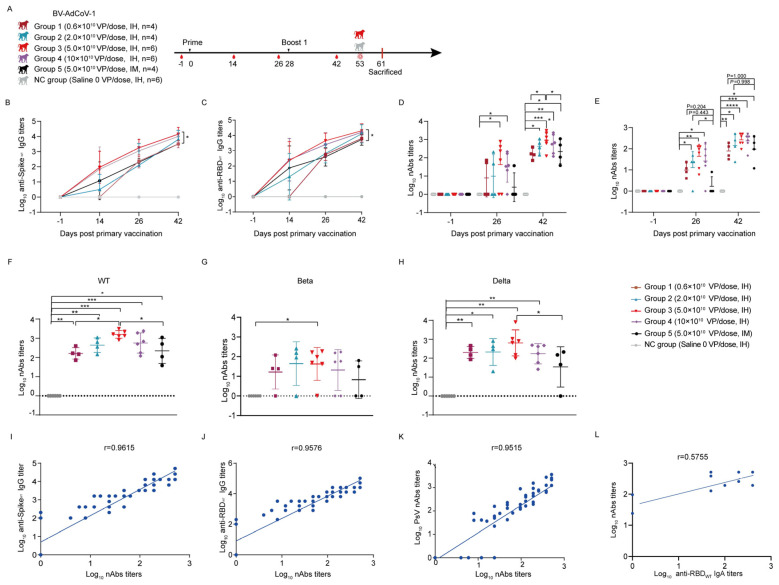
Aerosol inhalation of BV-AdCoV-1 elicits stronger antibody responses against SARS-CoV-2 in rhesus macaques than intramuscular injection. Thirty rhesus macaques (aged 3–4 years) were divided into six groups. Animals were vaccinated using a two–dose prime–boost regimen 28 days apart. Negative group: six rhesus macaques were vaccinated with saline. Group 1: four rhesus macaques were immunized by inhalation of BV-AdCoV-1 at a dose of 0.6 × 10^10^ VPs. Group 2: four rhesus macaques were immunized by inhalation with BV-AdCoV-1 at a dose of 2 × 10^10^ VPs. Group 3: six rhesus macaques were vaccinated by inhalation with BV-AdCoV-1 at the dose of 5 × 10^10^ VPs. Group 4: six rhesus macaques were vaccinated by inhalation with BV-AdCoV-1 at a dose of 10 × 10^10^ VPs. Group 5: four rhesus macaques were injected intramuscularly with BV-AdCoV-1 at a dose of 5 × 10^10^ VPs. (**A**) Schedule of vaccine administration and blood collection. Dosing (black triangles), terminal blood collections (red drop shapes), challenge (red crown–like structures) were carried out at the times. (**B**) Serum–binding IgG antibody titers against S of WT SARS-CoV-2 strain were measured on D–1, D14, D26, and D42 after primary vaccination by indirect ELISA. (**C**) Serum–binding IgG antibody titers against RBD of WT SARS-CoV-2 strain were measured on D–1, D14, D26, and D42 after primary vaccination by indirect ELISA. (**D**) Serum pseudovirus neutralizing antibody titers against WT SARS-CoV-2 on D–1, D26, and D42 after primary vaccination. (**E**) Serum neutralizing antibody titers against authentic WT strain (GD108# strain), on D–1, D26 post–primary vaccination, and D42, 14 days after the booster immunization. (**F**) Serum neutralizing antibody titers against pseudovirus of WT strain on D42. (**G**) Serum neutralizing antibody titers against pseudovirus of Beta strain on D42. (**H**) Serum neutralizing antibody titers against pseudovirus of Delta strain on D42. (**I**) Correlation between authentic neutralizing antibody titers against authentic WT strain (GD108# strain) and anti–S IgG antibody titers on D26 and D42. (**J**) Correlation between neutralizing antibody titers against authentic WT strain (GD108# strain) and anti–RBD IgG antibody titers on D26 and D42. (**K**) Correlation between neutralizing antibody titers against authentic WT strain (GD108# strain) and pseudovirus neutralizing antibodies against WT strain on D26 and D42. (**L**) Correlation between neutralizing antibody titers against authentic WT strain (GD108# strain) and binding IgA antibody titers against RBD of WT strain on D26 and D42. Data are presented as means ± SD. The lower limit of detection is a pseudovirus–neutralizing antibody titer of 20 in sera. When the value is below 20, it is calculated as 1. Statistical analysis was performed using two–way ANOVA, Tukey’s multiple comparison test, and two–sided Spearman rank correlation. * *p* < 0.05, ** *p* < 0.01, *** *p* < 0.001, **** *p* < 0.0001.

**Figure 4 vaccines-11-01427-f004:**
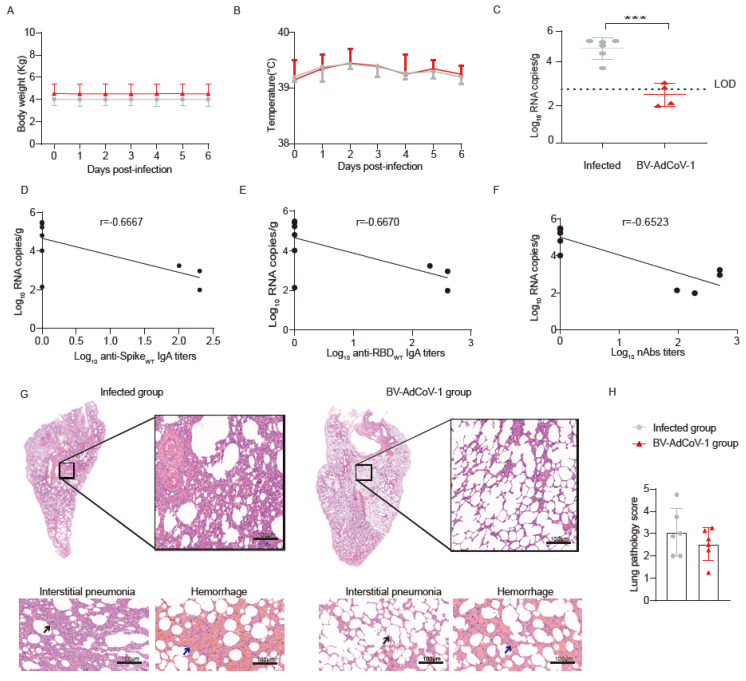
Immunoprotection of rhesus macaques vaccinated with BV-AdCoV-1 against SARS-CoV-2challenge. Rhesus macaques (*n* = 6/group) vaccinated either with two doses of aerosolized BV-AdCoV-1 (5 × 10^10^ VP/dose) or Saline (0 VP/dose) were challenged intratracheally and intranasally with 1 × 10^6^ pfu live SARS-CoV-2 (GD108# strain which is similar to the Wuhan–Hu–1 strain) on D25 post–Boost 1 vaccination. (**A**) Body weight monitored from dpi 0 to dpi 7. (**B**) Body temperature recorded from dpi 0 to dpi 7. (**C**) Viral load in lung tissues determined by RT–qPCR on dpi 7. (**D**) Correlation between serum binding IgA antibody titers against S of WT strain and the viral load in the lung tissues of the rhesus macaques in Group 3. (**E**) Correlation between serum binding IgA antibody titers against RBD of WT strain on D42 and viral load in the lung tissues of the rhesus macaques in Group 3. (**F**) Correlation between neutralizing antibody titers against authentic WT strain (GD108# strain) on D42 and viral load in the lung after challenge. (**G**) Histopathology of lung tissues by HE staining. Lung injuries of the non–vaccinated and vaccinated macaques were evaluated 7 days after the challenge. Blinded evaluation and histopathological scoring were performed by a pathologist. The black arrow indicates thickened alveolar wall. Arrows show signs of interstitial pneumonia, thickened alveolar septa, infiltration of inflammatory cells around the alveolar cavity and pulmonary septa (black arrows), and hemorrhage (blue arrows). Interstitial pneumonia is characterized by thickened alveolar septa and infiltration of inflammatory cells in the interstitial tissues. (HE 10×, scale bar = 100 µm). (**H**) Lung pathology score of HE stained samples. Pathological changes included thickening of alveolar walls and septa, infiltration of inflammatory cells, and diffuse lung hemorrhage. Scores were recorded according to all lesions across all lobes for each macaque and plotted. Dpi: day post–infection. Data are presented as means ± SD. Statistical analysis was performed using two–way ANOVA, Tukey’s multiple comparison test, and two–sided Spearman rank correlation. *** *p* < 0.001.

**Figure 5 vaccines-11-01427-f005:**
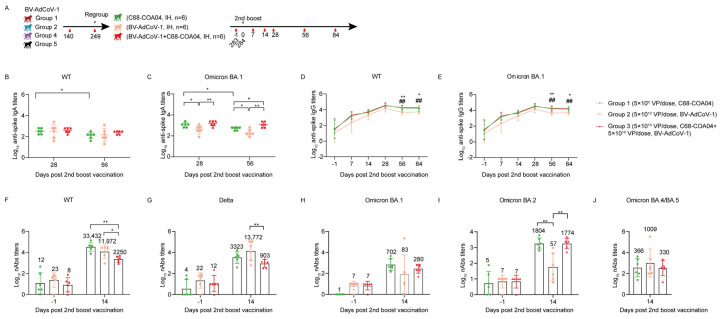
Homologous and heterologous booster vaccines enhance neutralizing antibody titers against WT and Omicron subvariants. Eighteen rhesus macaques (aged 3–4 years) from Groups 1, 2, 4, and 5 were regrouped into BV-AdCoV-1, C68–COA04, and C68–COA04 plus BV-AdCoV-1 groups (*n* = 6/group). (**A**) Experimental study schedule for the second booster vaccine. (**B**) Serum binding IgA antibody titers against S of WT strain in rhesus macaques on D28 and D56 determined by indirect ELISA. (**C**) Serum binding IgA antibody titers against S of Omicron BA.1 strain in rhesus macaques on D28 and D56 determined by indirect ELISA. (**D**) Serum binding IgG antibody titers against S of WT strain in rhesus macaques determined by indirect ELISA. (**E**) Serum binding IgG antibody titers against S of Omicron BA.1 strain in rhesus macaques determined by indirect ELISA. (**F**) Neutralizing antibody titers of immune sera against WT strain pseudovirus. (**G**) Neutralizing antibody titers of immune sera against Delta strain pseudovirus. (**H**) Neutralizing antibody titers of immune sera against Omicron BA.1 strain pseudovirus. (**I**) Neutralizing antibody titers of immune sera against Omicron BA.2 strain pseudovirus. (**J**) Neutralizing antibody titers of immune sera against Omicron BA.4/5 strain pseudovirus. Data are presented as means ± SD. Data are presented as means ± SD. The lower limit of detection is an IgA titer of 20 in sera. The lower limit of detection is a pseudovirus–neutralizing antibody titer of 20 in sera. The lower limit of detection is an IgG titer of 100 in sera. When the value is below the lower limit of detection, it is calculated as 1. Data are presented as means ± SD. Statistical analysis was performed using two–way ANOVA. * *p* < 0.05, ** *p* < 0.01, ^##^ *p* < 0.01.

## Data Availability

The authors confirm that the data supporting the findings of this study are available within the article and its Appendix A.

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
