# Peer review of "Aerosol Inhalation of Chimpanzee Adenovirus Vectors (ChAd68) Expressing Ancestral or Omicron BA.1 Stabilized Pre–Fusion Spike Glycoproteins Protects Non–Human Primates against SARS-CoV-2 Infection"

_vaccines, 2023, doi:10.3390/vaccines11091427_

Round 1
Reviewer 1 Report
The manuscript under title (Aerosol inhalation of chimpanzee adenovirus vectors (ChAd68) expressing ancestral or Omicron BA.1 stabilized pre-fusion spike glycoproteins protects non-human primates against SARS-CoV-2 infection) discusses very interesting topic (vaccines against SARS-CoV-2). It is well written and constructed. The topic is relevant and interesting, and the authors have good experience in this area. The application of vaccine by aerosol other than intranasal is superior and easy to apply.
I suggest to authors divide the part of (2.6. Immunogenicity Study Design in Rhesus Macaques) into two parts: vaccination and challenge and the other part measuring durability of the antibody response after a second booster.
Further control that should be considered is to challenge rhesus macaques after 249 weeks from the first vaccination to determine the protective efficacy after this period.
The figures are appropriate except the histological figs are (for me) of low quality and I think this is referred to the journal-specified area for figures.
Author Response
Dear Collaborative Peer Review Team,
Thank you for your valuable comments and advice. First of all, we would like to express our sincere gratitude to the reviewers for their constructive and positive comments. Those comments have been very helpful for revising and improving our manuscript. We have reviewed them carefully and accordingly made corrections which we hope will meet your approval. The revised sections are highlighted in red in the paper. The main corrections and specific responses to each reviewer’s comments are the followings:
Point 1: I suggest to authors divide the part of (2.6. Immunogenicity Study Design in Rhesus Macaques) into two parts: vaccination and challenge and the other part measuring durability of the antibody response after a second booster.
Response 1: Thanks for your helpful suggestion. We have made this modification according to your suggestion, including 2.6.1 Vaccination and challenge and 2.6.2 Durability of antibody response and sequential immunization. The revised sections are highlighted in red in the paper.
Point 2: Further control that should be considered is to challenge rhesus macaques after 249 weeks from the first vaccination to determine the protective efficacy after this period.
Response 2: Thanks for your thoughtful suggestion. ABSL-3 laboratories with experience in challenge of rhesus macaques are scant, and very expensive. It is now well accepted that a strong correlation between S- and RBD-specific binding antibody titers at D249 with serum neutralizing activity that we have observed and discussed in our study is a correlate of protection (KIZZMEKIA S. CORBETT, et al. Immune correlates of protection by mRNA-1273 vaccine against SARS-CoV-2 in nonhuman primates. Science. Vol 373, Issue 6561, 2021). So, short of a challenge study, we think that we can still reasonably infer that the vaccine has protective efficacy. In any case, the remaining monkeys were re-immunized at D284 and thus would no longer be available for challenge.
Point 3: The figures are appropriate except the histological figs are (for me) of low quality and I think this is referred to the journal-specified area for figures.
Response 3: Thank you for your comment. We have checked the original figures and it is indeed due to space limitations.
Best wishes,
Ke Wu
Reviewer 2 Report
SARS-Cov-2 vaccines can potentially be more effective, provide broader immunity and may be more long lasting administered via attenuated adenovirus vectors via inhalation. The current study was done in cynomolgus monkeys and rhesus macaques. Application of the vaccine by inhalation has the advantage of limiting transmission of the virus, because elicited immunity is stronger in the nose and lungs, which are the routes of infection. Intramuscular and intranasal injections are not as effective as applications by inhalation. Also, doses of vaccine might be reduced when applied via aerosol. This reviewer considers the paper to be of high quality and well-presented. Data are clearly presented and comprehensive.
This reviewer would prefer Figure 2 to be moved into the Results section and just referred to in Methods (i.e., see Figure 2, Results).
Some minor corrections:
775: SRAS-CoV-2 change
856: corrections needed
Could the “In summary…” paragraph at the end of the Discussion be separated into a Conclusions section? The Discussion is highly detailed.
Would the inhalation route of administration be useful for other common vaccines? Which ones?
If this approach were used in humans, would a combination of inhalation and intramuscular inoculation be most effective? How and when might these approaches be utilized in humans? What are the barriers to human application?
Author Response
Dear Collaborative Peer Review Team,
Thank you for your valuable comments and advice. First of all, we would like to express our sincere gratitude to the reviewers for their constructive and positive comments. Those comments have been very helpful for revising and improving our manuscript. We have reviewed them carefully and accordingly made corrections which we hope will meet your approval. The revised sections are highlighted in red in the paper. The main corrections and specific responses to each reviewer’s comments are the followings:
Point 1: This reviewer would prefer Figure 2 to be moved into the Results section and just referred to in Methods (i.e., see Figure 2, Results).
Response 1: Thanks for your helpful suggestion. We have made this modification according to your suggestion. The revised sections are highlighted in red in the paper. Please see Line 439 - Line 461.
Some minor corrections:
Point 2: 775: SRAS-CoV-2 change
Response 2: Thanks for your correction. We have made this modification. The revised sections are highlighted in red in the paper. Please see Line 767.
Point 3: 856: corrections needed.
Response 3: Thanks for your correction. We have made this modification. The revised sections are highlighted in red in the paper. Please see Line 848.
Point 4: Could the “In summary…” paragraph at the end of the Discussion be separated into a Conclusions section? The Discussion is highly detailed.
Response 4: Thanks for your helpful suggestion. We have made this modification according to your suggestion. The revised sections are highlighted in red in the paper. Please see Line 901.
Point 5: Would the inhalation route of administration be useful for other common vaccines? Which ones?
Response 5: Thanks for your insightful suggestion. The inhalation route of administration can be useful for other common vaccines. Inhalable vaccines can activate the mucosal immune system by directly contacting the mucosal surface of the respiratory tract. This immune response can generate specific antibodies, activate and regulate immune cells such as T cells and B cells, providing protection against the virus. Besides, inhalable vaccines can activate antigen-presenting cells in the respiratory tract, such as dendritic cells and macrophages, triggering local innate immune responses. These immune responses activate T-cells, induce antibody production, and are responsible not only for strong local immune responses but also systemic immunization.
The aerosolised adenovirus type-5 vector-based COVID-19 vaccine (Ad5-nCoV) has been approved for emergency use by the WHO. An adenovirus-vectored tuberculosis vaccine is undergoing Phase 1 clinical trial (NCT02337270). HPV vaccine also explored the aerosol vaccination which exhibited serum antibody titers that were comparable to those induced by intramuscular and it could induce anti-HPV16 VLP IgA secreting cells in PBMC and SIgA in secretions (Denise Nardelli-Haefliger, et al. Immune responses induced by lower airway mucosal immunisation with a human papillomavirus type 16 virus-like particle vaccine. Vaccine, Volume 23, Issue 28, 25 May 2005, Pages 3634-3641). Aerosolization with adenovirus-based vaccines is also directly applicable to respiratory virus vaccines, in particular RSV and parainfluenza virus vaccines.
Point 6: If this approach were used in humans, would a combination of inhalation and intramuscular inoculation be most effective? How and when might these approaches be utilized in humans? What are the barriers to human application?
Response 6: Thank you for your insightful suggestion. We think that a combination of inhalation and intramuscular inoculation will be very promising. As reported in aerosolised adenovirus type-5 vector-based COVID-19 vaccine (Ad5-nCoV) phase 1 clinical trial results (Shipo Wu, et al. Safety, tolerability, and immunogenicity of an aerosolised adenovirus type-5 vector-based COVID-19 vaccine (Ad5-nCoV) in adults: preliminary report of an open-label and randomised phase 1 clinical trial. The Lancet Infectious Diseases, VOLUME 21, ISSUE 12, P1654-1664, 2021), the mixed vaccination group received an initial intramuscular (5 × 1010 viral particles) vaccine on day 0, followed by an aerosolized booster (2 × 1010 viral particles) vaccine on day 28 (MIX group) which induced the highest anti-SARS-CoV-2 spike receptor IgG antibody and SARS-CoV-2 neutralizing antibody geometric mean titers compared with two aerosol groups and intramuscular groups.
In the future, more studies are needed to explore the benefits of this combination method in a large population.
Although there are no obvious regulatory barriers for human application, current COVID vaccines are highly cross-protective and very large efficacy trials would be needed to demonstrate the superiority of the vaccine combination when most of the population has been already infected or vaccinated.
Best wishes,
Ke Wu